biomechanics

animal movement, jumping, insects, ballistics, leap angle, robotic exploration

**Author for correspondence:**
Alberto Vailati
e-mail: alberto.vailati@unimi.it

# Optimal leap angle of legged and legless insects in a landscape of uniformly distributed random obstacles

Fabio Giavazzi[1], Samuele Spini[2], Marina Carpineti[2] and Alberto Vailati[2]

[1]Dipartimento di Biotecnologie Mediche e Medicina Traslazionale, and [2]Dipartimento di Fisica A. Pontremoli, Università degli Studi di Milano, 20133 Milano, Italy

FG, 0000-0003-4930-0592; SS, 0000-0002-5592-6367; MC, 0000-0002-8766-703X; AV, 0000-0002-3119-6021

We investigate theoretically the ballistic motion of small legged insects and legless larvae after a jump. Notwithstanding their completely different morphologies and jumping strategies, some legged and legless animals have convergently evolved to jump with a take-off angle of 60°, which differs significantly from the leap angle of 45° that allows reaching maximum range. We show that in the presence of uniformly distributed random obstacles the probability of a successful jump is directly proportional to the area under the trajectory. In the presence of negligible air drag, the probability is maximized by a take-off angle of 60°. The numerical calculation of the trajectories shows that they are significantly affected by air drag, but the maximum probability of a successful jump still occurs for a take-off angle of 59–60° in a wide range of the dimensionless Reynolds and Froude numbers that control the process. We discuss the implications of our results for the exploration of unknown environments such as planets and disaster scenarios by using jumping robots.

## 1. Introduction

Some animal species have evolved the ability of jumping, both as a fast locomotion method and as an escape manoeuver [1]. Compared to deambulation or locomotion by crawling, jumping allows achieving a fast-displacement by simultaneously overcoming natural obstacles distributed across the landscape [2]. The evolutive process leading to the optimization of the jumping performances is strongly influenced by the features of the habitat [3], and by the ecology of the species, which both contribute to determining the selection of a particular take-off angle.

**Table 1.** Kinematic parameters of legged and legless jumpers.

| species | mass (×10$^{-6}$ kg) | body length (×10$^{-3}$ m) | effective diameter (×10$^{-3}$ m) | take-off speed (m s$^{-1}$) | take-off angle | substrate | ref. |
|---|---|---|---|---|---|---|---|
| gall midge larva (*Asphondylia* sp.) | 1.27 | 3.28 | 1.2 | 0.85 | 63.3 $\pm$ 3° | plastic | [8] |
| fruit fly larva (*Ceratitis capitata*) | 17 | 8.5 | 2.8 | 1.17 | 60° | — | [9] |
| froghopper (*Philaenus spumarius*) | 12.3 | 6 | 4 | 4 | 58 $\pm$ 2.6° | — | [10] |
| | | | | | 53.2 $\pm$ 13° | epoxy | [7] |
| | | | | | 53.6 $\pm$ 14° | ivy leaves | [7] |

Jumping performances become remarkable in insects that use a catapult mechanism to amplify their muscular power and achieve long-range jumps by storing elastic energy inside their exoskeleton [4]. When the elastic energy is suddenly released by unlocking a latch, the body is projected into the air at high speed at distances significantly larger than the size of their body [5]. Insects have devised drastically different methods to jump by using a catapult mechanism. Froghoppers (*Philaenus spumarius*) take advantage of elastic deformation of their chitinous exoskeleton to achieve a rapid extension of their hind legs [6]. As a result, the body undergoes an acceleration as large as 4000 m s$^{-2}$ and enters a ballistic phase with top speeds up to 4 m s$^{-1}$. The presence of legs allows froghoppers to select take-off angles in the range $36° < \theta < 88°$. The mean angle at take-off is $53° \pm 14°$ [7] (table 1), significantly larger than the take-off angle of 45° needed to achieve maximum range.

Jumping driven by a catapult mechanism can also be achieved in the absence of highly specialized limbs like those present in froghoppers. A remarkable example is represented by the legless jumping of the larvae of the Mediterranean fruit fly (*Ceratitis capitata*). In this species, eggs are deposited under the skin of fruit, and when they hatch the larvae eat the host fruit, and abandon it when they are ready to pupate into the soil. The transfer to soil represents the critical moment when larvae are most vulnerable to predators. To minimize risks, larvae can move quickly by using a peculiar hydrostatic catapult mechanism that allows them to jump to distances of the order of 12 cm, namely more than 10 times their body length, in a fraction of a second [9]. In this species, a hydrostatic skeleton made by flexible outer muscular bands is anchored to a flexible skin layer, which confines an inner fluid region [11]. To jump, the larva first contracts its longitudinal ventral muscles to form a loop and anchors its head to the tail by using a pair of mouth hooks. The inner pressure of the body is then increased by contraction of helical muscular bands, and the sudden release of the mouth hooks gives rise to an abrupt straightening of the body, leading to a rapid acceleration phase ending with take-off at a velocity of about 1.2 m s$^{-1}$ and a take-off angle of 60° [9] (table 1).

A similar hydrostatic catapult mechanism is adopted by the larvae of the gall midge (*Asphondylia* sp.) [8]. In this case, latching is achieved by micrometre-scale finger-like microstructures distributed across the width of body segments. The sudden release of this latch leads to a take-off velocity of 0.85 m s$^{-1}$ at a take-off angle of $63° \pm 3°$ (table 1).

Notwithstanding their drastically different morphologies, froghoppers and the larvae of the fruit fly and gall midge have convergently evolved to jump with a take-off angle close to 60°. The understanding of the reasons behind this peculiar choice could shed new light on the selective pressure exerted by the geometrical and statistical features of the environment, and drive the development of bioinspired robotic devices suitable for efficient exploration of territories with unknown features. Moreover, froghoppers represent the key vectors for the *Xylella fastidiosa* Wells bacterium, which led to a dramatic epidemic disease on olive trees in the Mediterranean area [12]. The dispersal capabilities of froghoppers are still not very well known and a recent field study has shown that they could be more effective than expected [13]. A deeper understanding of the key factors that lead to the effectiveness of dispersion of froghoppers could help the identification of strategies to mitigate the fast spread of the dieback of olive trees.

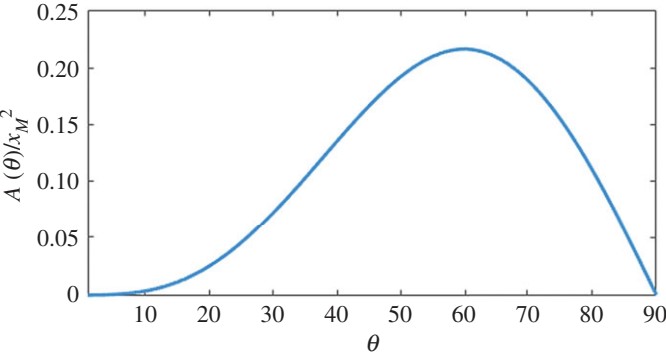

**Figure 1.** Area below the trajectory as a function of take-off angle normalized by the square of the maximum range $x_M$. The area exhibits an absolute maximum at $\theta = 60°$.

In this work, we show that a take-off angle of 60° maximizes the probability of overcoming obstacles of random size and position scattered across the landscape. We solve numerically the dimensionless equations that describe the kinematic motion of the animal in the presence of air drag and show that the features of the motion are completely determined by the dimensionless Reynolds and Froude numbers at take-off. We demonstrate that a take-off angle of 60° maximizes the probability of a successful jump in a very wide region of the parameter space, largely encompassing the conditions of interest for the jump of froghoppers and the larvae of fruit fly and gall midge.

## 2. Optimal strategy for jumping over a random obstacle

Living beings like froghoppers and the larvae of fruit fly and gall midge have a typical size of the order of a fraction of 1 cm (table 1) and are surrounded by obstacles whose size can largely exceed theirs. Under these conditions, both the maximum height of the jump and its range become important in overcoming an obstacle. A leap angle of 60° represents a good compromise between jump height and range because it maximizes their product or, which is the same, the area below the trajectory. To show this, let us assume that the take-off of the animal occurs at a fixed velocity $v_0$, and is only affected by the gravitational acceleration $g$, in the absence of air friction. Under these conditions, the motion completely occurs in a plane perpendicular to the ground and the equations of motion $\ddot{x} = 0$ and $\ddot{y} = -g$ can be easily integrated to yield the time evolution of the components of the displacement from the initial position:

$$x(t) = v_0 \cos(\theta) t \tag{2.1}$$

and

$$y(t) = v_0 \sin(\theta) t - \frac{1}{2} gt^2, \tag{2.2}$$

where $x$ and $y$ are the horizontal and vertical displacements, respectively, and $\theta$ is the take-off angle. The range of the jump, i.e. the distance travelled along the horizontal direction before landing at $y = 0$, is:

$$x_R(\theta) = 2 x_M \sin(\theta) \cos(\theta), \tag{2.3}$$

where $x_M$ is the maximum range achieved for a take-off angle of 45°:

$$x_M = \frac{v_0^2}{g}. \tag{2.4}$$

By integrating the trajectory across $x$ from the initial position $x = 0$ to the range of the jump $x_R$ the area below the curve as a function of $\theta$ (figure 1) is:

$$A(\theta) = \int_0^{x_R} y(x) \, dx = \frac{2}{3} x_M^2 \sin^3(\theta) \cos(\theta) \tag{2.5}$$

and derivation of equation (2.5) with respect to $\theta$ immediately shows that the area has an absolute maximum for $\theta = 60°$. From equation (2.5), one can appreciate that the area below the trajectory is

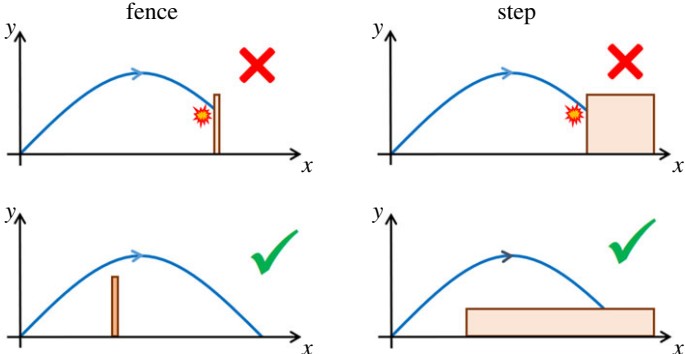

**Figure 2.** Success of a jump. Obstacles can be either represented by vertical fences (left column) or steps (right column). If the top-left edge of the obstacle is located above the trajectory the jump fails (top row), while when it is below the jump is successful (bottom row).

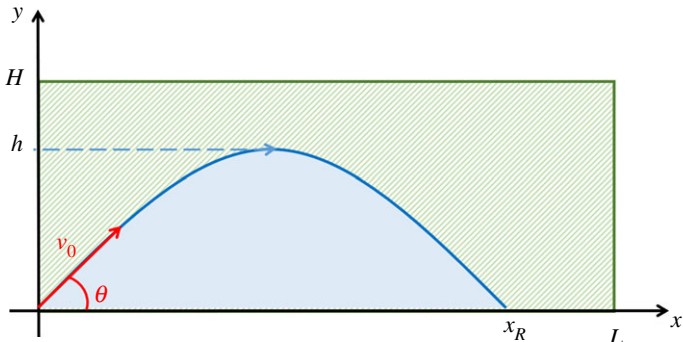

**Figure 3.** Probability of a successful jump. The solid light-blue region below the trajectory marks the regions of possible positions of the top edge of the obstacle in a successful jump, while the green rectangle all the possible positions.

proportional to the product between the range (equation (2.3)) and top height $h(\theta) = x_M \sin^2(\theta)/2$ of the jump.

Although the maximization of the area below the trajectory optimizes both the range and the maximum height of the jump, one might argue that this combination of the two parameters is arbitrary and does not lead necessarily to optimal performances in a natural landscape. Conversely, we will demonstrate that in a landscape made of obstacles of random size and position, the probability of overcoming an obstacle with a jump is directly proportional to the area below the trajectory. Let us assume that (i) the landscape is populated by obstacles, which can be either vertical fences (figure 2, left column), or steps (figure 2, right column); (ii) a jump is considered not successful when the trajectory passes below the top left edge of an obstacle (figure 2, top row), and successful when it passes above it (figure 2, bottom row); (iii) the height of the obstacles is uniformly distributed in the range $[0, H]$, where $H > x_M/2$, and their position is uniformly distributed in the range $[0, L]$, where $L > x_M$; and (iv) the initial velocity $v_0$ of the jump is fixed.

Owing to the uniform distribution of height and position of the obstacles, the probability $P(\theta)$ of a successful jump can be calculated directly as the area of the region of possible positions of the top-left edge of the obstacle below the trajectory (figure 3, solid light-blue area below the trajectory), normalized by the area of the region of all the possible positions (figure 3, rectangle of base $L$ and height $H$):

$$P(\theta) = \frac{A(\theta)}{HL}. \tag{2.6}$$

Equation (2.6) shows that the area $A(\theta)$ below the trajectory represents a direct measure of the probability of a successful jump and, in combination with equation (2.5), that the maximum probability is achieved for a take-off angle of 60° (figure 1).

# 3. Effect of air friction

The results derived in the previous section are obtained under the implicit hypothesis of negligible air friction. However, as discussed by Vogel [14], the bio-ballistics of small projectiles like the insects listed in table 1 is strongly affected by air drag, which becomes increasingly important as the size of the projectile is diminished. Under such circumstances, the presence of air drag can determine a significant decrease of both the range and height of the jump and in turn of the area below the trajectory. Indeed, equation (2.5) for the area below the trajectory has been obtained in the absence of air friction. We will show that, although air drag affects significantly the trajectories of the insects listed in table 1, the optimal take-off angle is affected only marginally by its presence. One firm result of the model reported above is represented by equation (2.6), which states that the probability of a successful jump is proportional to the area under the trajectory. This result is valid under generic conditions, both in the presence and in the absence of air drag. Therefore, to calculate the probability of a successful jump under realistic conditions one just needs to determine the area below the trajectories in the presence of air drag. The effect of inertia and viscous drag on the body are combined into the dimensionless Reynolds number:

$$Re = \frac{\rho l v}{\eta}, \tag{3.1}$$

where $\rho$ and $\eta$ are, respectively, the density of air (1.2 kg m$^{-3}$ at 20°C) and its shear viscosity ($1.8 \times 10^{-5}$ Pa s), and $l$ is a typical effective diameter of the body (table 1). When $Re < 1$ the motion of the body occurs in the Stokes regime, and the drag force is proportional to $v$, while for $Re > 1000$ the flow occurs in the Newton regime, and the drag force becomes proportional to $v^2$. Following Vogel [14], we introduce a drag coefficient that takes into account the transition between these regimes:

$$C_d(Re) = \frac{24}{Re} + \frac{6}{1 + Re^{1/2}} + 0.4. \tag{3.2}$$

The modulus of the drag force acting of the body can be calculated as

$$D = \frac{1}{2} C_d \rho S v^2, \tag{3.3}$$

where $S = \pi(l/2)^2$ is the cross sectional area that the body offers to the air flow. Following Landau & Lifschitz [15], gravitational effects can be accounted for by introducing the dimensionless Froude number:

$$Fr = \frac{v^2}{l g'}, \tag{3.4}$$

where $g' = g(\rho_p - \rho)/\rho$ is the reduced acceleration of gravity [16] and $\rho_p$ is density of the insect.

The equations of motion can be written in terms of the dimensionless variables $(\tilde{x}, \tilde{y}) = (x, y)/x_M$ and $\tilde{t} = t/t_0$, where $t_0 = v_0/g$ is a characteristic time of flight:

$$\ddot{\tilde{x}} = -\mu \dot{\tilde{x}} \tag{3.5}$$

and

$$\ddot{\tilde{y}} = -\mu \dot{\tilde{y}} - 1, \tag{3.6}$$

where

$$\mu = \frac{3}{4} C_d(Re_0 \tilde{v}) Fr_0 \tilde{v} \tag{3.7}$$

is a dimensionless drag coefficient that only depends on the dimensionless velocity $\tilde{v}$ and on the Reynolds and Froude numbers at take-off, defined by $Re_0 = \rho l \, v_0/\eta$ and $Fr_0 = v_0^2/lg'$, respectively. According to equations (3.2) and (3.7), for small $Re_0$, the drag coefficient attains a constant value $\mu_0 \sim 18 \, Fr_0/Re_0$ and equations (3.5) and (3.6) can be solved analytically. For larger $Re_0$, the nonlinear dependence of $\mu$ from $\tilde{v}$ makes the equation hard to solve analytically, but solutions can be still found numerically. We integrated the equations numerically by using Euler's method detailed in [17] to obtain the area under the trajectory as a function of take-off angle for a set of parameters mirroring the three representative cases of leaping insects and larvae detailed in table 1. The numerically calculated dimensionless areas $\tilde{A}(\theta)$ are shown in figure 4, where dashed and dotted lines represent

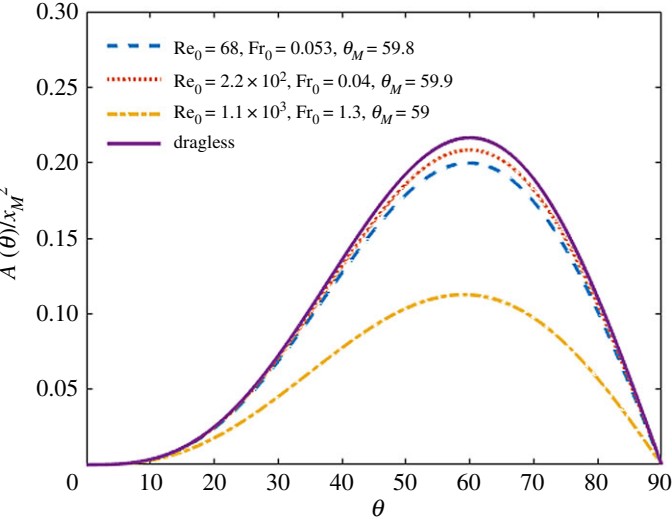

**Figure 4.** Numerically calculated area below the trajectories in the presence of air drag normalized by the square of the maximum range $x_M$. Dashed line: gall midge; dotted line: fruit fly larva; dashed-dotted line: froghopper. The parameters used to process the trajectories mirror those reported in table 1. The solid line represents the area in the absence of air drag, calculated from equation (2.5).

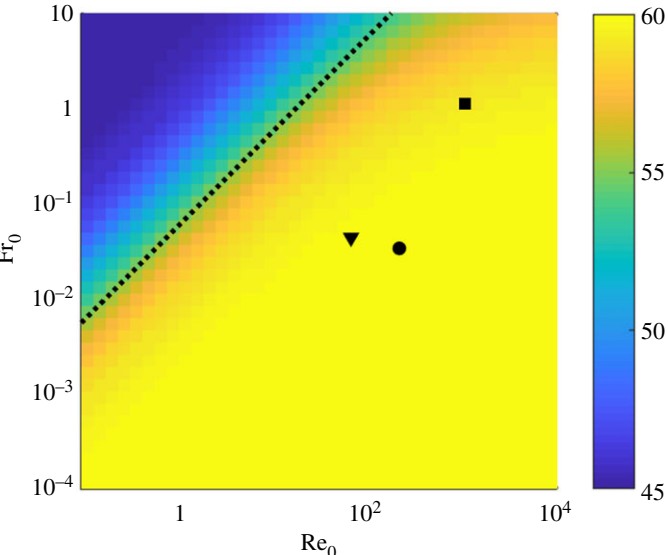

**Figure 5.** Optimal leap angle as a function of the Reynolds and Froude numbers at take-off. Black symbols: gall midge (downward triangle); fruit fly larva (circle); froghopper (square). The dotted line corresponds to $Fr_0 = Re_0/18$, which is equivalent to the condition $\mu = 1$ in the low Reynolds number limit (see equations (3.2) and (3.7)).

results obtained in the presence of air drag, while the solid line those obtained without air drag. One can appreciate that, although air drag determines a decrease of $\tilde{A}(\theta)$, the maxima of the curves are located in a narrow range of angles $59° < \theta < 59.9°$, very close to the value $\theta = 60°$ predicted analytically in the absence of air drag.

To assess the robustness of these results, we systematically investigated the optimal leap angle, corresponding to the maximum of $\tilde{A}(\theta)$, over a wide range of the two control parameters $Re_0$ and $Fr_0$. As shown in figure 5, the parameters of all the jumpers considered in this work fall well within a large domain of the parameter space where the optimal leap angle is almost indistinguishable from the drag-free ideal case $\theta = 60°$. For small Reynolds numbers ($Re \lesssim 10^2$), this domain is identified by the simple condition $\mu \ll 1$, which corresponds to $Fr_0 \ll Re_0/18$ (dotted line in figure 5).

# 4. Discussion

A jump angle of 60° is peculiar of the insect species that we have analysed, but does not represent a general feature of legged and legless jumping insects. In general, jumping insects either make directed jumps to pass obstacles, to hit prey and hosts, or jump regardless of obstacles to evade predators. A remarkable example is represented by hedgehog fleas (*Archaeopsyllus erinacei*), which adopt a take-off angle in the range $28° < \theta < 52°$, with a mean angle of $39° \pm 6°$ [18]. By putting the parameters for the jump of hedgehog fleas in our model one finds that the optimal jump angle is still very close to 60°. This apparent contradiction can be solved by taking into account that our model determines the best take-off angle needed to trespass random obstacles, but some animal species do not jump to achieve this task. Indeed, in the case of hedgehog fleas an important role of the jumping behaviour is to land on a moving host [19]. To achieve this task a flea aims at maximizing the probability of hitting a target, and a forward jump with a take-off angle of about 40° represents a better choice than a jump directed more vertically.

We have demonstrated that the fact that a take-off angle close to $\theta = 60°$ maximizes the probability of success of a jump in the presence of uniformly distributed random obstacles represents a robust result that does not depend on the details of the model adopted. This remarkable result can be profitably used to optimize the features of autonomous robots used for the exploration of environments populated by unknown obstacles, such as other planets [20], or Earth regions than can be dangerous for human beings, like nuclear disaster sites or earthquake scenarios [21,22]. A well-known approach is to use rover robots that usually have large size and mass, move by using wheels or tracks, and navigate thanks to a high degree of technology and a closed-loop control of their movements. Although these characteristics allow a fine control of the navigation, they accomplish a limited capability of mapping extended territories. A different approach is the use of colonies of small and agile robots, typically tens [23], with simple design and functions and open-loop control, which can map extended territories. As the size of robots decreases, they likely have to overcome obstacles whose size is comparable or even larger than their own one [24,25]. A bio-mimetic approach suggests that jumping has a great potentiality of success [26–28] in rough terrains. In fact, in recent years a large amount of research has focused on the refinement of miniaturized robots inspired by jumping organisms, like, for example, froghoppers [29], locusts [30], insects [27,31,32] or even soft worms [33,34]. The final choices made in the design of a jumping robot result from a complex balance among constraints connected to take-off, air flight and landing. Flight issues have often to do with posture adjustment, landing with stability and take-off with the mechanisms of energy storing and fast conversion in kinetic energy for jumping [29,30,33]. To our knowledge, the probability of success in overcoming an obstacle of random size and position has not been considered yet during the design process, and the take-off angle is often chosen *a priori* [33,35].

The results reported in this work could inspire a different approach in the design of miniaturized robots, which could be particularly effective for the challenging case of groups of small robots that collectively explore a rough terrain. Under these conditions, the implementation of a robot hardwired to jump at an angle of 60° would allow attaining optimal performances in the exploration of unknown rough regions, sided by an extremely simple conceptual design.

Data accessibility. The Matlab code used for the numerical integration of the equations of motion in the presence of air drag is provided as the electronic supplementary material.

Authors' contributions. F.G. designed the study, developed the model, performed numerical analysis and helped draft the manuscript; S.S. conceived the study, contributed to the development of the model and critically revised the manuscript; M.C. participated in the design of the study and helped draft the manuscript; A.V. conceived the study, designed the study, coordinated the study, contributed to the development of the model, performed numerical analysis and drafted the manuscript. All authors gave final approval for publication and agree to be held accountable for the work performed therein.

Competing interests. We declare we have no competing interests.

Funding. We received no funding for this study.

Acknowledgements. We thank Marcello Re for valuable comments and suggestions.

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
