## [Peer Review File · Royal Society Open Science]

Review History

RSOS-202279.R0 (Original submission)

Review form: Reviewer 1

Is the manuscript scientifically sound in its present form?

Yes

Are the interpretations and conclusions justified by the results?

Yes

Is the language acceptable?

Yes

Do you have any ethical concerns with this paper?

No

Have you any concerns about statistical analyses in this paper?

No

Recommendation?

Accept as is

Comments to the Author(s)

Firstly, let me begin by saying that this is just plain an interesting and well done set of modelling work. I enjoyed it a lot, and was pulled into the question and love how the authors looked at the answer. I have several small suggestions, which the authors are welcome to take on board (or not) as they see fit - given that I was so interested in how they presented the data and what they did here, it would be churlish for me to mandate any changes. So take these as suggestions.

1) Figure 1, you may want to put fleas (Sutton et al., 2011) on this figure. Fleas never jump at 60 degrees (figure 8 from this paper), and that may be because they exist (with their smaller size and lower velocity) close to your 45 degree area on this plot. I'm not sure where the flea would fit on this plot, so if it causes things to get ugly, feel free not to include them.

2) Comment that froghoppers often jump at lower angles (Goetzke et al) than 60 degrees, but not quite at 45 derees (Figure 2 of this paper). The paper as written implies that the Goetzke hoppers jumped at 60 degrees, and while they sometimes did, the means are lower than that).

But, let me say, I liked this paper a lot. thank you!

Review form: Reviewer 2**Is the manuscript scientifically sound in its present form?**

Yes

Are the interpretations and conclusions justified by the results?

Yes

Is the language acceptable?

Yes

Do you have any ethical concerns with this paper?

No

Have you any concerns about statistical analyses in this paper?

No

Recommendation?

Accept as is

Comments to the Author(s)

This is a nice clear contribution showing why a jump angle of 60 degrees is optimal to pass random obstacles. I do not have major issues with the analysis or conclusion but I would like to point out that jump angle is not always 60 degrees in legged and legless jumping insects. This is predominantly because very few insects jump to pass random obstacles. Rather, insects either make directed jumps to pass obstacles or jump regardless of obstacles to evade predators. Nevertheless, the relevance of the optimal jumping angle for random obstacle passing in miniature robots is made very clear in the study.

Decision letter (RSOS-202279.R0)

Dear Miss Vailati

On behalf of the Editors, we are pleased to inform you that your Manuscript RSOS-202279 "Optimal leap angle of legged and legless insects in a landscape of uniformly-distributed random obstacles" has been accepted for publication in Royal Society Open Science subject to minor revision in accordance with the referees' reports. Please find the referees' comments along with any feedback from the Editors below my signature.

Please submit your revised manuscript and required files (see below) no later than 7 days from today's (ie 12-Apr-2021) date. Note: the ScholarOne system will 'lock' if submission of the revision is attempted 7 or more days after the deadline. If you do not think you will be able to meet this deadline please contact the editorial office immediately.

on behalf of Pietro Cicuta (Subject Editor)
openscience@royalsociety.org

Associate Editor Comments to Author:
Comments to the Author:

Please accept our apologies for the unusual delay in completing review of the paper, but two reviewers are broadly in favour of acceptance of your work, though there are a couple of minor comments that we would encourage you to address before submitting a final version of the paper.

Reviewer comments to Author:

Reviewer: 1

Comments to the Author(s)

Firstly, let me begin by saying that this is just plain an interesting and well done set of modelling work. I enjoyed it a lot, and was pulled into the question and love how the authors looked at the answer. I have several small suggestions, which the authors are welcome to take on board (or not) as they see fit - given that I was so interested in how they presented the data and what they did here, it would be churlish for me to mandate any changes. So take these as suggestions.

1) Figure 1, you may want to put fleas (Sutton et al., 2011) on this figure. Fleas never jump at 60 degrees (figure 8 from this paper), and that may be because they exist (with their smaller size and lower velocity) close to your 45 degree area on this plot. I'm not sure where the flea would fit on this plot, so if it causes things to get ugly, feel free not to include them.

2) Comment that froghoppers often jump at lower angles (Goetzke et al) than 60 degrees, but not quite at 45 derees (Figure 2 of this paper). The paper as written implies that the Goetzke hoppers jumped at 60 degrees, and while they sometimes did, the means are lower than that).

But, let me say, I liked this paper a lot. thank you!

Reviewer: 2

Comments to the Author(s)

This is a nice clear contribution showing why a jump angle of 60 degrees is optimal to pass random obstacles. I do not have major issues with the analysis or conclusion but I would like to point out that jump angle is not always 60 degrees in legged and legless jumping insects. This is predominantly because very few insects jump to pass random obstacles. Rather, insects either make directed jumps to pass obstacles or jump regardless of obstacles to evade predators. Nevertheless, the relevance of the optimal jumping angle for random obstacle passing in miniature robots is made very clear in the study.

===PREPARING YOUR MANUSCRIPT===

===PREPARING YOUR REVISION IN SCHOLARONE===

Author's Response to Decision Letter for (RSOS-202279.R0)

See Appendix A.

Decision letter (RSOS-202279.R1)

Dear Dr Vailati,

It is a pleasure to accept your manuscript entitled "Optimal leap angle of legged and legless insects in a landscape of uniformly-distributed random obstacles" in its current form for publication in Royal Society Open Science.

on behalf of Prof Pietro Cicuta (Subject Editor)
openscience@royalsociety.org

Appendix A

Dear Editor,

we thank the Reviewers for their appreciative remarks and comments. We revised the manuscript by taking into account all their requests and, as a result, it is apparent to us that the clarity of our paper has improved. We provide below a detailed list of changes made to the manuscript in response to the recommendations of the Reviewers.

Kind regards

Alberto Vailati

Reviewer: 1

Comments to the Author(s)

Firstly, let me begin by saying that this is just plain an interesting and well done set of modelling work. I enjoyed it a lot, and was pulled into the question and love how the authors looked at the answer. I have several small suggestions, which the authors are welcome to take on board (or not) as they see fit - given that I was so interested in how they presented the data and what they did here, it would be churlish for me to mandate any changes. So take these as suggestions.

We thank the Reviewer for his/her enthusiastic comments. We enjoyed doing this work and we are glad that he/she liked reading it.

1) Figure 1, you may want to put fleas (Sutton et al., 2011) on this figure. Fleas never jump at 60 degrees (figure 8 from this paper), and that may be because they exist (with their smaller size and lower velocity) close to your 45 degree area on this plot. I'm not sure where the flea would fit on this plot, so if it causes things to get ugly, feel free not to include them.

We have determined the optimal jump angle of hedgehog fleas by using our model, and the optimal jump angle is still of the order of 60°. This result is in apparent contradiction with the experimental distribution of take-off angles reported in Sutton et al, 2011. We have added a paragraph at the beginning of the “discussion” section, where we suggest a solution to this contradiction:

*“A remarkable example is represented by hedgehog fleas (*Archaeopsyllus erinacei*), which adopt a take-off angle in the range $28^\circ < \theta < 52^\circ$ with a mean angle of $39^\circ \pm 6^\circ$ (Sutton2011). By putting the parameters for the jump of hedgehog fleas in our model one finds that the optimal jump angle is still close to 60°. This apparent contradiction can be solved by taking into account that our model determines the best take-off angle needed to trespass random obstacles, but some animal species do not jump to achieve this task. Indeed, in the case of hedgehog fleas an important role of the jumping behavior is to land on a moving host (Burrows2009). To achieve this task a flea aims at maximizing the probability of hitting a target, and a forward jump with take-off angle of about 40° represents a better choice than a jump directed more vertically.”*

2) Comment that froghoppers often jump at lower angles (Goetzke et al) than 60 degrees, but not quite at 45 derees (Figure 2 of this paper). The paper as written implies that the Goetzke hoppers jumped at 60 degrees, and while they sometimes did, the means are lower than that).

We have revised the text to provide a more detailed description of the results by Goetze et al.:

“The presence of legs allows froghoppers to select take off angles in the range $36^\circ < \theta < 88^\circ$. The mean angle at take-off is of the order of $53^\circ \pm 14^\circ$ (Goetzke2019) (Table \ref{tab:jump}), significantly larger than the take-off angle of 45° needed to achieve maximum range.”

But, let me say, I liked this paper a lot. thank you!

Thank You!

Reviewer: 2

Comments to the Author(s)

This is a nice clear contribution showing why a jump angle of 60 degrees is optimal to pass random obstacles.

We thank the Reviewer for his/her appreciation of our work.

I do not have major issues with the analysis or conclusion but I would like to point out that jump angle is not always 60 degrees in legged and legless jumping insects. This is predominantly because very few insects jump to pass random obstacles. Rather, insects either make directed jumps to pass obstacles or jump regardless of obstacles to evade predators.

We have revised the manuscript to convey more clearly the fact that not all insects jump at an angle of 60°. More in detail:

- *We have revised a sentence in the abstract, that now reads: “**some** legged and legless animals have convergently evolved to jump with a take-off angle of 60°”*
- *We have revised the description of the jump in the larvae of fruit fly to detail the peculiar conditions where the jumping behavior is observed during the development of the organism:
“In this species, eggs are deposited under the skin of fruit, and when they hatch the larvae eat the host fruit, and abandon it when they are ready to pupate into the soil. The transfer to soil represents the critical moment when larvae are most vulnerable to predators. To minimize risks, larvae can move quickly by using a peculiar hydrostatic catapult mechanism that allows them to jump to distances of the order of 12 cm, namely more than ten times their body length, in a fraction of a second \cite{Maitland1992}.”*
- *We have added some sentences at the beginning of the discussion section where we state that not all legged and legless insects jump to avoid an obstacle:
“A jump angle of 60° is peculiar of the insect species that we have analyzed, but does not represent a general feature of legged and legless jumping insects. In general, jumping insects either make directed jumps to pass obstacles, to hit preys and hosts, or jump regardless of obstacles to evade predators. A remarkable example is represented by hedgehog fleas (*Archaeopsyllus erinacei*), which adopt a take-off angle in the range $28^\circ < \theta < 52^\circ$ with a mean angle of $39^\circ \pm 6^\circ$ \cite{Sutton2011}. By putting the parameters for the jump of hedgehog fleas in our model one finds that the optimal jump angle is still 60°. This apparent contradiction can be solved by taking into account that our model determines the best take-off angle needed to trespass random obstacles, but some animal species do not jump to achieve this task. Indeed, in the case of hedgehog fleas an important role of the jumping behavior is to land on a moving host \cite{Burrows2009}. To achieve this task a flea aims at maximizing the probability of hitting a target, and a forward jump with take-off angle of about 40° represents a better choice than a jump directed more vertically.”*

Nevertheless, the relevance of the optimal jumping angle for random obstacle passing in miniature robots is made very clear in the study.

We appreciate very much that the Reviewer has taken this opportunity to outline the strengths of our work.